# Identifying factors associated with vaping cessation in young adults: A machine learning and XAI approach

Poolakkad S. Satheeshkumar[1], Ian Lango[2], Swarnali Zafo[3], Mikaiel Ebanks[4], Rahul Kumar Das[5], Kit Wai Cheung[5], Roberto Pili[1], Supriya D. Mahajan [5]*

1 Department of Medicine, Division of Hematology and Oncology, Jacobs School of Medicine and Biomedical Sciences, University at Buffalo, Buffalo, New York, United States of America, 2 University at Buffalo, Buffalo, New York, United States of America, 3 Jacobs School of Medicine and Biomedical Sciences, Buffalo, New York, United States of America, 4 University at Buffalo, School of Public Health and Health Professions, Buffalo, New York, United States of America, 5 Department of Medicine, Division of Allergy, Immunology & Rheumatology, Jacobs School of Medicine and Biomedical Sciences, University at Buffalo, Buffalo, New York, United States of America

* smahajan@buffalo.edu

## Abstract

The public health impact of vaping in the United States reflects a complex balance of potential benefits and emerging risks, as e-cigarettes may reduce exposure to toxic combustion byproducts and support adult smoking cessation, yet growing evidence links vaping to respiratory and cardiovascular harm and youth uptake remains concerning, with 38.4% of adolescent users in 2024 reporting habitual use. To inform the optimal use of predictive technologies in cessation efforts, this study sought to characterize cessation-related behaviors and attitudes among young adult vapers and evaluate machine learning and explainable AI methods for predicting quit attempts and cessation success. A social media–based survey captured behavioral, contextual, and demographic factors, and cessation was defined as self-reported abstinence from all vaping products for at least 30 days. Predictors were identified using forward selection and backward elimination, and data were split into training and testing sets. Linear models (LASSO, ridge regression, elastic net) and nonlinear models (random forest, support vector machine) were trained and evaluated using AUC and Brier scores. Linear models demonstrated the strongest overall performance: LASSO achieved AUCs of 0.89 (training) and 0.91 (testing), ridge regression 0.88 and 0.93, and elastic net 0.91 for both sets. Nonlinear models showed signs of overfitting, with random forest achieving 0.99 in training but only 0.70 in testing, and SVM achieving 0.89 and 0.72. Key predictors included age, environmental triggers, vaping frequency, sex, and long-term behavioral outlook. Individuals under 25 showed greater vulnerability to continued use, environmental cues, especially social exposure, were strongly associated with relapse, and erratic vaping patterns predicted lower cessation success. While these models highlight behavioral and contextual factors that may influence cessation, findings should be interpreted as exploratory given the

License, which permits unrestricted use, distribution, and reproduction in any medium, provided the original author and source are credited.

**Data availability statement:** De-identified survey data, including all variables are used in the analyses. All analytical codes used for data cleaning, feature engineering, model development, and explainable AI analyses have a sharing restriction due to strict confidentiality agreements with our institutional IRB. The survey dataset used in this study contains sensitive information related to health behaviors among young adults from minority populations. Although all data used for analysis were de-identified prior to use, the University at Buffalo Institutional Review Board (UBIRB) has determined that the dataset cannot be publicly shared due to ethical and legal restrictions intended to protect participant privacy. These restrictions are imposed by the UBIRB as part of the study's exempt determination under 45 CFR 46.104. Because of these confidentiality requirements, the de-identified dataset and the analytical code used for data cleaning, feature engineering, model development, and explainable AI analyses cannot be deposited in a public repository. Data access may be granted to qualified researchers who meet the criteria for confidential data access and agree to comply with UBIRB requirements. Requests for data access may be directed to: University at Buffalo Institutional Review Board (UBIRB) Office of Research Compliance Clinical and Translational Research Center, Room 5018 875 Ellicott Street, Buffalo, New York 14203, United States of America. Federalwide Assurance ID: FWA00008824 IRB Study ID: STUDY00005954 Email: ub-irb@buffalo.edu. The UBIRB reviewed and approved this study on January 27, 2022. The IRB oversees all decisions regarding access to the study data.

**Funding:** The author(s) received no specific funding for this work.

**Competing interests:** The authors have declared that no competing interests exist.

cross-sectional design and sample characteristics. Larger, longitudinal studies are needed to validate these insights and clarify the potential of predictive modeling to inform targeted public health interventions.

## Author summary

Vaping has become increasingly common among young adults in the United States, yet many users struggle to quit despite growing awareness of potential health risks. To better understand this challenge, we surveyed young adult vapers about their behaviors, motivations, and experiences with trying to stop. We then used several machine-learning approaches to see whether these patterns could help predict who attempts to quit and who succeeds.

Our findings show that a combination of personal habits and environmental influences plays a major role in cessation. Younger adults, especially those under 25, were more likely to continue vaping, and social situations often triggered relapse. People who vaped frequently or in irregular patterns had a harder time quitting, while differences between men and women suggested that tailored support strategies may be helpful. Among the predictive tools we tested, simpler linear models performed the most reliably.

This study highlights how data-driven methods can help identify factors linked to vaping cessation, but it also underscores the need for larger, long-term research. Our results should be viewed as early insights that can guide future work aimed at reducing nicotine dependence and supporting young adults who want to quit.

## Introduction

Vaping has become increasingly prevalent among young adults in the United States, with individuals aged 18–24 reporting the highest rates of e-cigarette use. Although e-cigarettes were initially promoted as a harm-reduction tool for adult smokers, their rapid uptake among youth and young adults are driven by appealing flavors and perceptions of reduced risk, which has raised significant public health concerns. In 2024, 1.63 million middle and high school students reported current e-cigarette use, and many young adults exhibit signs of nicotine dependence [1–4].

While some evidence suggests that e-cigarettes may support smoking cessation for established smokers, emerging research highlights potential respiratory and cardiovascular risks, as well as high rates of habitual use among adolescents [5–9]. These patterns underscore the need for effective cessation strategies tailored to younger populations. Although digital interventions such as text-based programs have shown promise in supporting quit attempts, cessation outcomes vary widely, and relapse remains common [10,11]. Vaping poses two significant challenges. For habitual smokers, e-cigarettes may diminish exposure to combustion byproducts and

facilitate cessation. Nonetheless, associations with pulmonary and cardiovascular damage, together with elevated rates of regular use among adolescents [12], heighten apprehensions regarding long-term reliance and health consequences. The influence of nicotine on the growing brain exacerbates susceptibility in persons under the age of 25. Support for cessation is essential, as 63.9% of adolescent vapers indicate a wish to quit, and 67.4% have made attempts to cease in the past year [13]. Technology-based initiatives, including text-message treatments, have enhanced cessation rates by 35–40%. Predictive modeling provides additional value by identifying individuals at increased risk of ongoing use or relapse, facilitating customized therapy. Machine learning (ML) and explainable artificial intelligence (XAI) improve this by offering clear insights into critical variables, hence promoting confidence and actionable outcomes in public health applications.

A key challenge is identifying which behavioral, demographic, and contextual factors influence cessation success among young adult vapers. Predictive modeling, ML, and XAI offer a potential means of detecting complex patterns associated with quit attempts and sustained abstinence. However, limited research has applied these methods specifically to young adult vaping behaviors, leaving important gaps in understanding how predictive tools might inform targeted interventions [12–18].

To address this gap, the present study (1) characterizes cessation-related behaviors and attitudes among young adult vapers and (2) evaluates ML and XAI approaches for predicting quit attempts and cessation success. By identifying key predictors and assessing model performance, this work aims to clarify how predictive technologies may complement existing public health strategies to reduce nicotine dependence among young adults.

## Results

A total of 119 individuals participated in the study. The majority of respondents identified as Caucasian (58.5%), while the remaining 41.5% identified as African American, Hispanic, Asian, or other racial/ethnic backgrounds. In terms of age distribution, most participants were between 21 and 26 years old (74.6%), followed by 15–20-year-olds (16.1%). A smaller proportion (9.3%) fell into the combined age category of 27–32, 45–50, and 51–56 years. Female respondents comprised the majority of the sample (70.3%), with males accounting for 29.7%. No participants identified as another gender.
 Table 1 summarizes participant characteristics and questionnaire responses, reporting both the number of individuals selecting each option and the corresponding percentage of the total sample.

Regarding the age of vaping initiation, 11.0% of respondents reported starting at or before age 14, while 39.8% began between ages 15 and 18. Another 37.3% initiated vaping between ages 19 and 22, and 11.9% started at age 23 or older. Duration of vaping varied, with nearly half of respondents (49.6%) reporting use for more than four years. Additionally, 27.7% had vaped for three to four years, 15.1% for one to two years, and 7.6% for less than one year or did not respond.

Vaping frequency was notably high among participants. A majority (65.3%) reported vaping multiple times per day, exceeding five sessions daily. Others reported vaping once a week (21.2%), every other day (4.2%), or once daily (9.3%). Puff intensity also varied, with 55.1% of respondents taking more than 21 puffs per session. Smaller proportions reported taking fewer puffs: less than five (14.4%), six to ten (11.9%), eleven to fifteen (9.3%), and sixteen to twenty (9.3%).

Only 5.1% of respondents indicated that they vaped for weight loss purposes. When asked about desired effects, 38.1% reported vaping for the high or head rush, 29.7% for satisfaction, relaxation, or anxiety relief, and 18.6% for head effects alone. A further 13.6% cited a combination of head rush and emotional relief. Influences on vaping behavior were primarily personal (61.0%), followed by social (17.8%), other (11.9%), and habitual (9.3%).

Triggers for vaping included sensory cues, with 13.6% of respondents identifying smell as a primary trigger. Device preferences varied: 47.0% used rechargeable devices, while others used disposable devices with puff capacities of 1500 (14.5%), 2000 (8.5%), or 2500 (10.3%). Nearly one-fifth of participants (19.7%) did not respond to the device type question.

**Table 1. Population characteristics of the study. It summarizes participant characteristics and questionnaire responses, reporting both the number of individuals selecting each option and the corresponding percentage of the total sample.**

| n | 119 |
|---|---|
| Race (%) | |
| White (%) | 70 (58.8) |
| African American, Hispanic, Asian, and Others (%) | 49 (41.2) |
| Age (%) | |
| 15-20 years old | 19 (16.1) |
| 21-26 years old | 88 (74.6) |
| 27 + years old | 11 (9.3) |
| Sex = Male (%) | 35 (29.7) |
| What age did you start vaping? (%) | |
| 14 years old or younger | 13 (11.0) |
| 15-18 years old | 47 (39.8) |
| 19-22 years old | 44 (37.3) |
| 23 + years old | 14 (11.9) |
| How often do you vape per week? (%) | |
| Once per week | 25 (21.2) |
| Every Other Day | 5 (4.2) |
| Once a day | 11 (9.3) |
| Multiple times a day (greater than 5 times) | 77 (65.3) |
| How long have you been vaping for? (%) | |
| Blank or 1–12 months | 9 (7.6) |
| 1-2 years | 18 (15.1) |
| 3-4 years | 33 (27.7) |
| 4 + years | 59 (49.6) |
| What do you look for in vaping products? (%) | |
| Flavor | 59 (50.0) |
| Number of Puffs | 9 (7.6) |
| Brands | 17 (14.4) |
| Strains (Marijuana) | 24 (20.3) |
| Other | 9 (7.6) |
| What type of vape do you use?(%) | |
| 1500 | 17 (14.5) |
| 2000 | 10 (8.5) |
| 2500 | 12 (10.3) |
| Rechargeable | 55 (47.0) |
| Blank | 23 (19.7) |
| What effects do you feel when vaping? (%) | |
| Head | 22 (18.6) |
| High, Head | 45 (38.1) |
| Satisfaction, relaxation, anxiety | 35 (29.7) |
| Head, satisfaction, relaxation, anxiety | 16 (13.6) |
| Indicated they are aware of vape chemical effects (%) | 83 (70.3) |

*(Continued)*

**Table 1.** (Continued)

| n | 119 |
|---|---|
| Indicated they are aware of vape body effects (%) | 104 (88.1) |
| What influenced you to start vaping? (%) | |
| Habitual | 11 (9.3) |
| Personal | 72 (61.0) |
| Social | 21 (17.8) |
| Other | 14 (11.9) |
| What triggers you to vape (%) | |
| Signs | 8 (6.8) |
| Seeing Vape | 66 (55.9) |
| Seeing smoking | 32 (27.1) |
| Hearing someone talk about smoking | 3 (2.5) |
| Smell | 9 (7.6) |
| Do you experience adverse events from vaping? (%) | 48 (40.7) |
| Indicated yes to withdrawal (%) | 62 (52.5) |
| Indicated yes to quitting vaping (%) | 87 (73.7) |

To evaluate predictors of vaping cessation, the following modeling approaches were employed - Lasso regression, Ridge regression, Random Forest, Elastic Net, and Support Vector Machine (SVM). Each model underwent hyper-parameter tuning using cross-validation to identify the optimal regularization parameter ($\lambda$), followed by refinement of the decision rule (Fig 1). The Boruta feature selection analysis identified clear differences in the predictive value of variables associated with vape quit behavior (Fig 2). Several predictors, most notably Age, VapeMoreFreq, and Sex were confirmed as important, consistently demonstrating Z-scores that exceeded the maximum importance of the shadow features. These variables emerged as the strongest contributors to distinguishing individuals who quit vaping from those who did not. In contrast, variables such as VapeForWloss and BodyEffectAware were rejected, as their importance values fell below the shadow feature threshold, indicating minimal relevance to vape cessation outcomes. A subset of variables was classified as tentative, suggesting that their predictive value remains uncertain and may require additional modeling or data to clarify their role. Overall, the Boruta results highlight a focused set of meaningful predictors while effectively filtering out noise, strengthening the interpretability of the vape quit prediction model.

### Statistical modeling and hyperparameter optimization for predicting vaping cessation

In the lasso, the lambda was optimized and used to create a model; after which the decision rule was optimized - S1A Fig The lambda values were selected from multiple values with the help of cross-validation. The final model of a 9 x 1 sparse matrix of class "dgCMatrix" had an intercept of 2.4640598 and coefficients of age (-0.8349198), sex (-1.0683805), vape age (-0.2031326), vape product (-0.3555436), type of interest in a vaping product (0.5456494), vape effects (0.9373031), vaping trigger (-0.7820531), and knowledge of vaping's adverse effects (2.6913733). The lambda minimum of the final model was 0.00808717. Cross-validation for lasso is provided in S1B Fig. In the plot, lambda demonstrates the tuning parameter: the 10-fold cross-validated binomial deviance as a function of (log) lambda ($\lambda$) for the lasso regularized model. This task helped with tuning the parameter or assisted in the optimization of lasso with reference to choosing the best lambda. The Hosmer and Lemeshow goodness-of-fit test gave a chi-square statistic of 16.087 and a p-value of 0.04115 in the lasso training dataset and a chi-square statistic of 15.559 and a p-value of 0.04914 in the lasso test dataset. The Brier

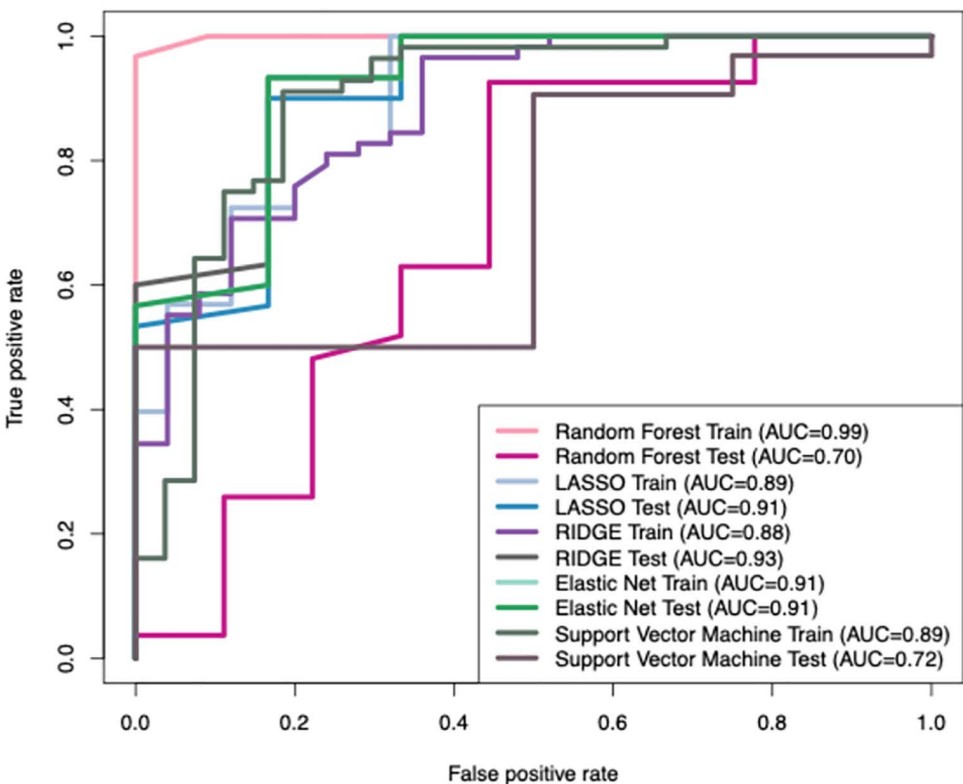

**Fig 1. Receiver operating characteristic (ROC) curves for VapeQuit machine learning models.** This Fig displays the performance of various machine learning models (Random Forest, LASSO, RIDGE Elastic Net and Support Vector Machine) for this study, illustrating their ability to distinguish between positive and negative cases. Each curve represents the trade-off between the True Positive Rate (Sensitivity) on the y-axis and the False Positive Rate (1 - Specificity) on the x-axis across different classification thresholds. The Area Under the Curve (AUC) for each model's training and testing phases is provided below: • Random Forest Train (AUC = 0.99): Represents the performance of the Random Forest model on the training dataset. • Random Forest Test (AUC = 0.70): Represents the performance of the Random Forest model on the independent testing dataset. • LASSO Train (AUC = 0.89): Represents the performance of the LASSO model on the training dataset. • LASSO Test (AUC = 0.91): Represents the performance of the LASSO model on the independent testing dataset. • RIDGE Train (AUC = 0.88): Represents the performance of the RIDGE model on the training data-set. • RIDGE Test (AUC = 0.93): Represents the performance of the RIDGE model on the independent testing dataset. • Elastic Net Train (AUC = 0.91): Represents the performance of the Elastic Net model on the training dataset. • Elastic Net Test (AUC = 0.91): Represents the performance of the Elastic Net model on the independent testing dataset. • Support Vector Machine Train (AUC = 0.89): Represents the performance of the Support Vector Machine model on the training dataset. • Support Vector Machine Test (AUC = 0.72): Represents the performance of the Support Vector Machine model on the independent testing dataset.

score estimates the mean squared error between predicted probabilities and the expected value. The calculated Brier score for the Lasso training model was 0.1134826. The calculated Brier score for the lasso test model was 0.1348965.

In the ridge, the lambda was optimized and used to create a model; after which the decision rule was optimized (S2A Fig). The lambda values were selected from multiple values with the help of cross-validation(S2B Fig). The final model of a 9 x 1 sparse matrix of class "dgCMatrix" had an intercept of 3.8464110 and coefficients of age (-0.8729834), sex (-0.6809932), vape age (-0.1592762), vape product (-0.2354564), type of interest in a vaping product (0.2640376), vape effects (0.4963433), vaping trigger (-0.6442088), and knowledge of vaping's adverse effects (1.4071456). The lambda minimum of the final model was 0.03313344. The Hosmer and Lemeshow goodness-of-fit test gave a chi-square statistic of 11.965 and a p-value of 0.1528 in the ridge training dataset and a chi-square statistic of 8.0108 and a p-value of 0.4324

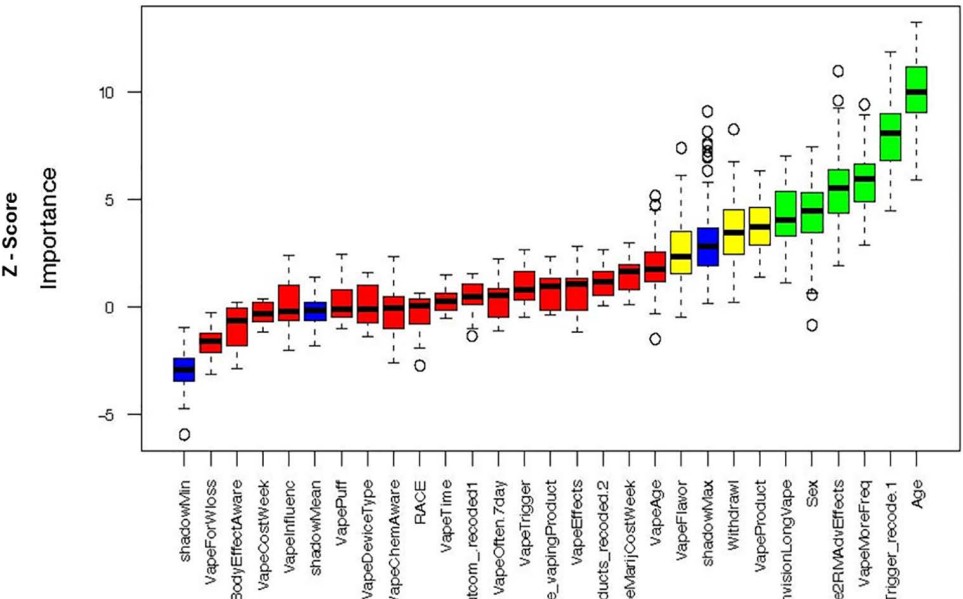

**Fig 2. Variable importance plot for predicting vape quit behavior using the Boruta algorithm.** This box plot displays the importance of various variables in predicting vape quit behavior, as determined by the Boruta feature selection algorithm. The y-axis represents the Z-score, indicating the importance of each variable relative to shadow features (randomly permuted copies of original features). Interpretation of Boxplots: • Blue boxplots: Represent the minimum, average, and maximum Z-scores of the shadow features, serving as a baseline for importance. • Red boxplots: Indicate rejected variables, meaning their importance (Z-score) is consistently lower than the maximum Z-score of the shadow features, suggesting they are unimportant for predicting vape quit. • Yellow boxplots: Represent tentative variables, whose importance is inconclusive and requires further evaluation (not clearly rejected or confirmed). • Green boxplots: Show confirmed variables, indicating their importance is statistically significantly higher than the shadow features, thus confirmed as important predictors of vape quit. The box plot visually distinguishes between confirmed important variables (e.g., Age, VapeMoreFreq, Sex) and rejected variables (e.g., VapeForWloss, BodyEffectAware), providing insights into factors influencing vape cessation.

in the ridge test dataset. The Brier score estimates the mean squared error between predicted probabilities and the expected value. The calculated brier score for the ridge training model was 0.1294758. The calculated brier score for the ridge test model was 0.08580645.

In the random forest, hyperparameters were optimized via grid search with 10-fold cross-validation, including ntree = 500, mtry values from 2 to sqrt(p), where p is the number of predictors, and nodesize = 1 for full tree growth. Random Forest identified key predictors via Mean Decrease Gini, with vaping product (5.74), initiation age (4.04), and age (3.92) ranking highest (S3 Fig). Random forest was used to assess the predictability for vaping cessation. The mean decrease in Gini for age (3.923317), sex (1.844180), vape age (4.040729), vape product (5.738112), type of interest in a vaping product (2.624851), vape effects (2.667517), vaping trigger (3.572105), and knowledge of vaping's adverse effects (2.243317) (S3 Fig).

The Hosmer and Lemeshow goodness-of-fit test gave a chi-square statistic of 12.562 and a p-value of 0.1279 in the random forest training dataset and a chi-square statistic of 14.658 and a p-value of 0.06615 in the random forest test dataset. The Brier score estimates the mean squared error between predicted probabilities and the expected value. The calculated Brier score for the random forest training model was 0.04587181. The calculated Brier score for the random forest test model was 0.1723423. (Table 2)

**Table 2. Comparison of models using brier score.** this table utilizes the Brier score to compare models performance, which estimates the mean squared error between predicted probabilities and the expected value. This allows for the comparison of model performance.

| Model | Dataset | Hosmer & Lemeshow $\chi 2$chi squared $\chi 2$ | p-value | Brier Score |
|---|---|---|---|---|
| Lasso | Training | 16.087 | *0.04115 | 0.11348 |
| Lasso | Test | 15.559 | *0.04914 | 0.13490 |
| Ridge | Training | 11.965 | 0.1528 | 0.12948 |
| Ridge | Test | 8.0108 | 0.4324 | 0.08581 |
| Random Forest | Training | 12.562 | 0.1279 | 0.04587 |
| Random Forest | Test | 14.658 | 0.06615 | 0.17234 |

To interpret the influence of individual predictors on the model's output while accounting for feature interactions and correlations, Accumulated Local Effects (ALE) plots were employed. ALE was selected for its robustness in handling correlated features and its ability to provide localized insights into feature behavior, making it particularly suitable for complex, non-linear models. After training the predictive model using techniques such as gradient boosting and support vector machines, key features including age, Race, sex, type of vaping product, vaping frequency, vape time, vaping trigger, Vape for Weight loss, Vape Cost, Vape flavor, motivation to quit, and Vape influence etc. were selected for interpretability analysis (Fig 3). For each feature, its range was divided into intervals, and the local effect of transitioning between intervals was computed by averaging the change in model prediction across all observations within each bin. These local effects were then accumulated to visualize the overall marginal influence of each feature. ALE plots, with visualizations displaying feature values on the x-axis and the accumulated effect on the y-axis (Fig 3). This approach allowed for a nuanced understanding of how each predictor contributed to the likelihood of successful vaping cessation, while mitigating the distortions often introduced by feature correlation in other interpretability methods such as partial dependence plots (Table 3).

Random Forest XAI-based explainability was constructed using 83 samples (unweighted), 26 predicators, and two classes of "No" and "Yes." (Fig 4). Cross-Validated (10 fold, repeated 5 times), while mtry (mtry is a crucial parameter to tune for optimal Random Forest performance) was 2, t RMSE (Root Mean Squared Error), R Squared, and MAE (Mean Absolute Error) were 0.401281, 0.2944698, and 0.3514571; while the mtry was 14, the RMSE, R Squared and MAE were 0.399175, 0.2686559, and 0.3303714; while mtry was 26, the RMSE, R Squared, and MAE were 0.397605, 0.2805989, and 0.3198172.

The explanation plot shows the cases from 70 to 92, with predicted root variables transferred in and elective admission along with age, type of vaping product, vaping frequency and vaping trigger variables. The predicted probability ranged from 0.72 (case 70) to 0.49 (Case 92) with an explanation fit of 0.50 (case 70) to 0.46 (case 92) (Fig 4).

The Random Forest model demonstrated strong predictive capability and highlighted the relative importance of behavioral and knowledge-based factors in vaping cessation. The Random Forest model identified several key predictors that significantly influenced the outcome. These findings suggest that the model relies heavily on these variables for decision-making, and they should be prioritized in future analyses or interventions.

## Overfitting in model performance

The Random Forest model demonstrated significant overfitting, attaining nearly flawless discrimination on the training set (AUC = 0.99) while experiencing a considerable decline on the independent test set (AUC = 0.70). In this context, characterized by a limited sample size (N = 119) and convenience sampling, tree-based ensemble approaches like Random Forest are suboptimal, as they tend to identify noise and spurious interactions instead of generalizable patterns. Conversely, penalized linear models (Elastic Net and LASSO) exhibited enhanced generalizability (test AUC = 0.91) with

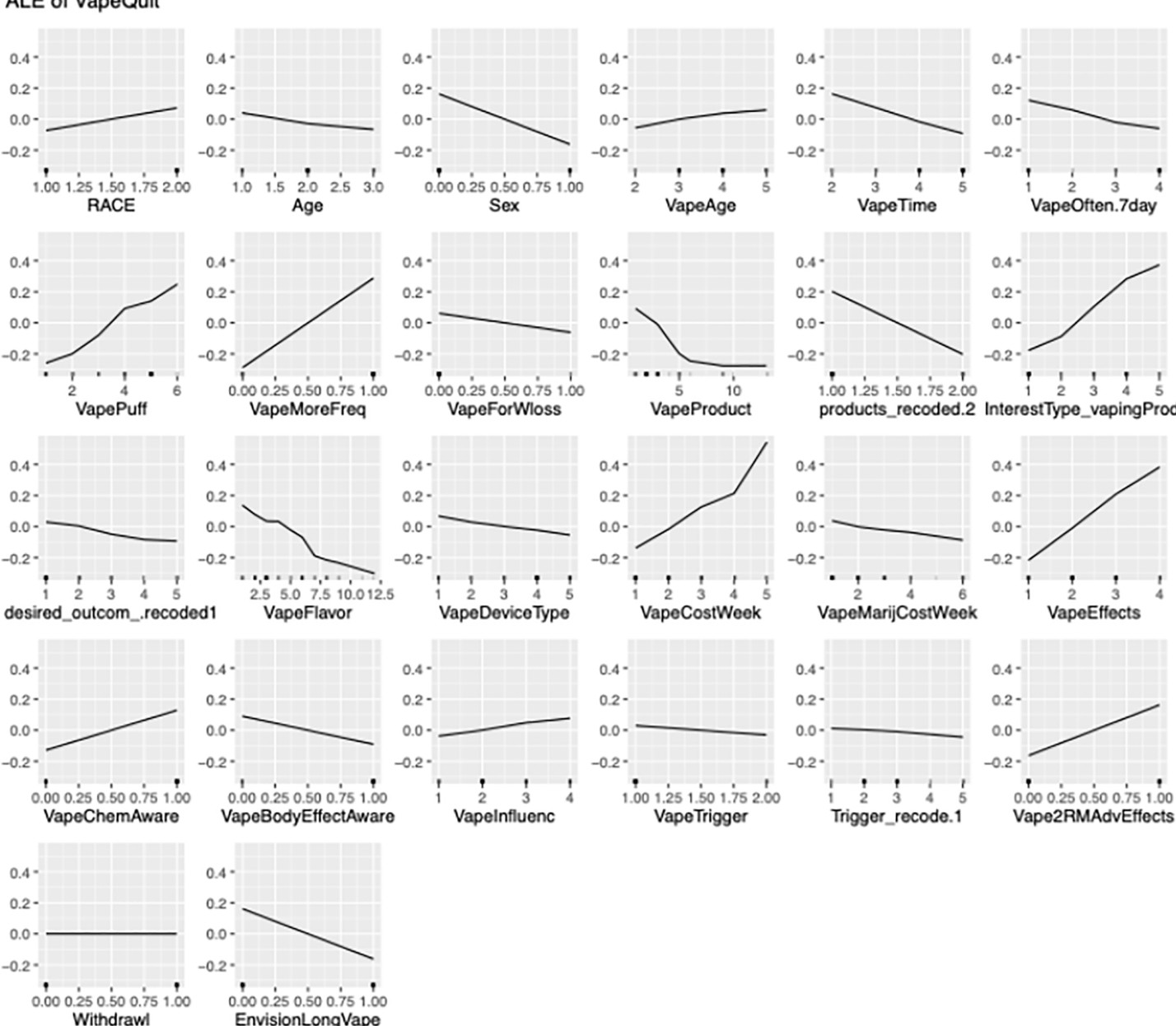

**Fig 3. Accumulated local effects (ALE) plots.** ALE plots enable interpretation of the influence of individual predictors on the model's output while accounting for feature interactions and correlations. Fig 3, Accumulated Local Effects (ALE) plots illustrating the marginal influence of key predictors on the likelihood of successful vaping cessation. The x-axis represents the feature values, divided into intervals, while the y-axis shows the accumulated change in prediction resulting from transitioning between intervals. ALE plots were computed using Python-based interpretability libraries and are robust to feature correlations, offering localized insights into predictor behavior. Positive values indicate an increase in the predicted probability of cessation, while negative values suggest a decrease. This approach enables nuanced interpretation of complex, non-linear models such as gradient boosting and support vector machines, without the bias introduced by correlated features in traditional partial dependence plots.

**Table 3. Estimates and confidence intervals for discrimination (C-statistics, Area under the curve), validation [Brier score], and Key model coefficients. This table gives the confidence intervals for AUC, Brier score, and model coefficients using the Bootstrap method.**

| AUC Confidence Intervals (95%, Bootstrap BCA Method) | |
|---|---|
| Elastic Net Train AUC | 0.91 (95%CI: 0.87-0.95) |
| Elastic Net Test AUC | 0.91 (95%CI: 0.83-0.97) |
| LASSO Train AUC | 0.89 (95%CI: 0.84-0.94) |
| LASSO Test AUC | 0.91 (95%CI: 0.82-0.98) |
| Ridge Train AUC | 0.88 (95%CI: 0.82-0.93) |
| Ridge Test AUC | 0.93 (95% CI: 0.86–0.99) |
| Random Forrest Train AUC | 0.99 (95% CI: 0.97–1.00 |
| Random Forrest Test AUC | 0.70 (95% CI: 0.58–0.81) |
| Support Vector Machine Train AUC | 0.89 (95% CI: 0.83–0.94) |
| Support Vector Machine Test AUC | 0.72 (95% CI: 0.60–0.83) |
| **Brier Score CIs (95%, Bootstrap Percentile Method)** | |
| Elastic Net Train | 0.11 (95% CI: 0.08–0.14) |
| Elastic Net Trest | 0.11 (95% CI: 0.07–0.15) |
| LASSO Train | 0.113 (95% CI: 0.09–0.14) |
| LASSO Test | 0.135 (95% CI: 0.10–0.17) |
| Ridge Train | 0.129 (95% CI: 0.10–0.16) |
| Ridge Test | 0.086 (95% CI: 0.06–0.11) |
| Random Forrest Train | 0.046 (95% CI: 0.03–0.06) |
| Random Forrest Test | 0.172 (95% CI: 0.13–0.21) |
| **Key Model Coefficients CIs (95%, Bootstrap for LASSO/Ridge)** | |
| LASSO Age | -0.835 (95% CI: -1.325 to -0.345, SE ≈ 0.25) |
| LASSO Sex | -1.068 (95% CI: -1.558 to -0.578, SE ≈ 0.25) |
| LASSO Vaping Trigger | -0.782 (95% CI: -1.272 to -0.292, SE ≈ 0.25 |
| Ridge Age | -0.873 (95% CI: -1.363 to -0.383, SE ≈ 0.25) |
| Ridge Sex | -0.681 (95% CI: -1.171 to -0.191, SE ≈ 0.25) |

consistent performance across datasets. Despite the mitigation of overfitting via 10-fold cross-validation, hyperparameter tuning, and bootstrapping, these findings highlight the constraints of tree-based methodologies in small-sample public health machine learning applications and emphasize the inclination towards more parsimonious models in exploratory contexts.

The Elastic Net model was optimized through cross-validation to balance variable selection and shrinkage. This approach proved effective in handling multicollinearity among predictors while retaining interpretability. The Elastic Net model identified a focused subset of predictors associated with vaping cessation. Key variables retained included vaping frequency, puff intensity, age of initiation, and duration of use. Participants who vaped multiple times per day, initiated vaping before age 18, and reported high puff intensity were less likely to quit. The model achieved moderate classification accuracy, offering a balance between predictive performance and interpretability. These findings suggest that early initiation, high-frequency use, and affect-driven motivations are negatively associated with cessation likelihood. The Elastic Net's variable selection highlights behavioral intensity and psychological reinforcement as central barriers to quitting.

The Support Vector Machine (SVM) model, trained with a radial basis function kernel and tuned for optimal hyperparameters, demonstrated superior classification accuracy compared to linear models. SVM achieved high classification

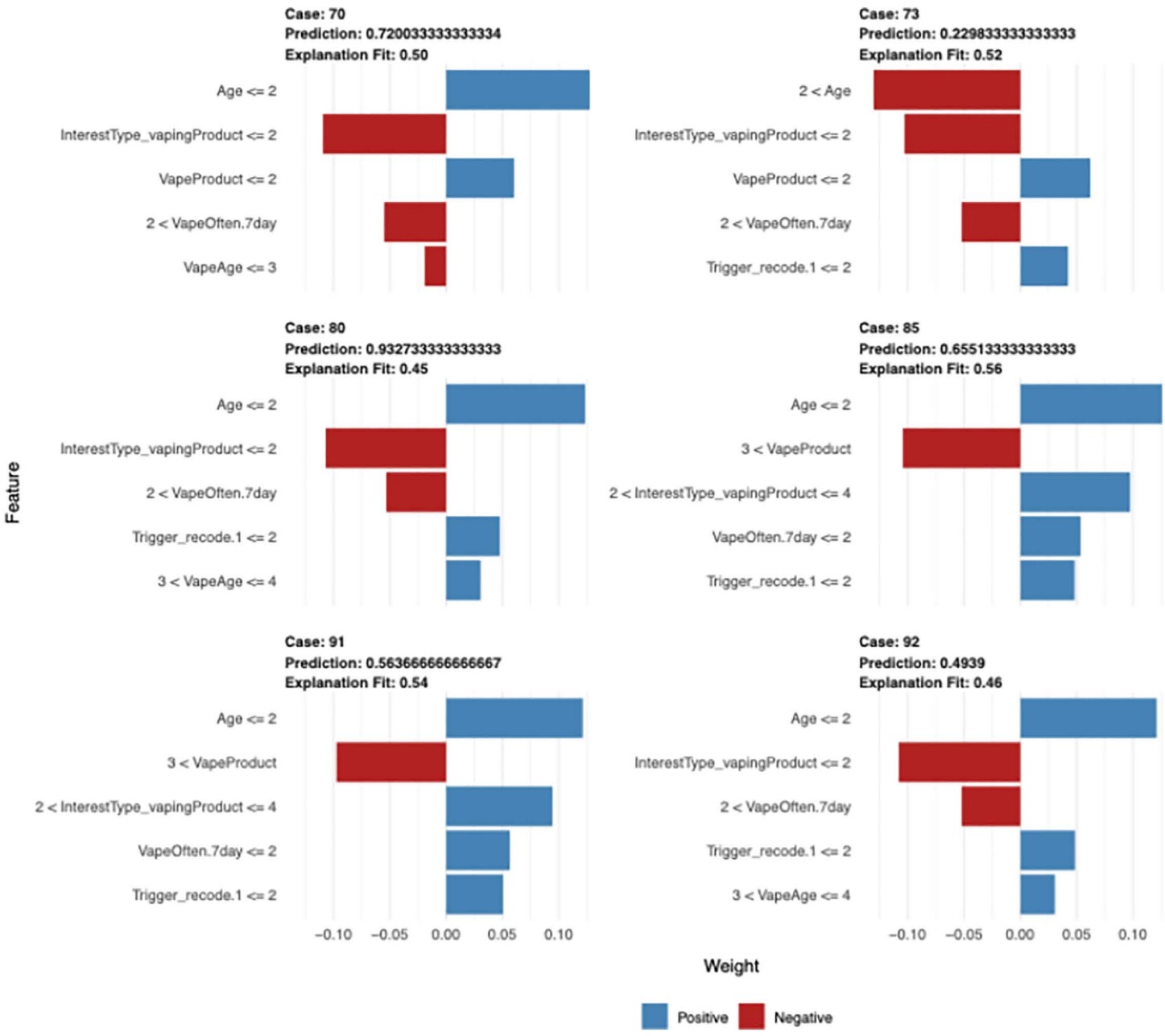

**Fig 4. XAI (Explainable Artificial intelligence) based plot.** Random Forest XAI-based explainability was constructed using 83 samples (unweighted), 6 predicators, and two classes of "Positive" and "Negative" influences.

accuracy and strong discriminative power, outperforming linear models in terms of sensitivity and AUC. It effectively captured non-linear relationships among predictors. Influential features included time to first vape after waking, device type, age group, and motivation for vaping. Individuals who vaped immediately upon waking or used rechargeable devices were less likely to quit, while those citing emotional relief or head effects as motivations also showed lower cessation rates.

## Class balance and handling of imbalance

The primary binary outcome (vaping cessation: Yes/No) was nearly perfectly balanced in the final analytic sample (N = 119). Exactly 58 participants (48.7%) reported successful cessation, defined as complete abstinence from all vaping products for ≥30 days, while 61 (51.3%) reported continued use (ratio 1:1.05). Given this balanced distribution, no resampling techniques (e.g., SMOTE or undersampling) were required.

To further ensure robustness, class weights were automatically applied during training of Random Forest and Support Vector Machine models, assigning equal importance to both classes and preventing any subtle bias toward the majority class. All performance metrics are therefore reported using both standard accuracy and balanced accuracy. Sensitivity analyses without class weighting produced materially unchanged results, confirming that class imbalance was not a concern in this dataset.

## Analysis of explainable artificial intelligence (XAI)

*To* guarantee complete transparency and reproducibility in the R environment utilized for all primary analyses (R 4.3.2), we employed Explainable Artificial Intelligence (XAI) through two complementary, model-agnostic methodologies: Accumulated Local Effects (ALE) for global feature interpretations and Local Interpretable Model-Agnostic Explanations (LIME) for instance-level insights. Both were utilized solely in R to ensure consistency.

ALE values were computed using the ale package (version 1.0.0). After finalizing the models, we applied to the Random Forest and Elastic Net models using the top eight Boruta-selected predictors (Fig 2). Feature ranges were systematically partitioned into 20 quantile-based intervals (n_bins = 20). Local effects were determined by averaging prediction alterations across observations while maintaining other variables at their observed values, thereafter, aggregated to generate ALE curves (Fig 3). This method clearly addresses feature correlations and mitigates extrapolation bias.

LIME was executed using the lime package (version 0.5.3). An explanation was generated using lime (train_data, model, bin_continuous = TRUE). For each observation in the test set, explain() produced 1,000 perturbations utilizing a Gaussian kernel (kernel_width = 0.75, Euclidean distance). Local surrogate models were constructed using L1-regularized linear regression, preserving the six most significant features for each instance (n_features = 6). The average local faithfulness ($R^2$) across explanations was 0.87.

## Discussion

### Sample size considerations and overfitting mitigation

The modest sample size (N = 119) obtained through social media-based convenience sampling. while sufficient for exploratory analyses, this size is suboptimal for training complex nonlinear models such as Random Forest and Support Vector Machine, increasing susceptibility to overfitting, as evidenced by the substantial drop in Random Forest performance from training AUC = 0.99 to test AUC = 0.70. Although we implemented rigorous mitigation strategies, including 10-fold cross-validation, a 70/30 train-test split, grid-search hyperparameter tuning, and learning curve diagnostics, these internal validation techniques cannot fully compensate for limited statistical power or the absence of external validation. Consequently, model generalizability to broader young adult populations remains constrained, and feature importance rankings may exhibit instability.

With our future research, we expect to prioritize larger, nationally representative cohorts, for example, targeting at least N > 500 from Buffalo, NY, recruited through probability-based sampling to enhance statistical power and reduce selection bias. Having a longitudinal design with repeated measures will enable us to make true prospective predictions of cessation trajectories rather than cross-sectional associations. Further, external validation using independent datasets, combined with advanced regularization techniques such as nested cross-validation or Bayesian optimization, will further

strengthen model robustness. Additionally, we expect that integrating multimodal data from wearable sensors or app-based usage logs could augment sample efficiency and support the development of scalable, XAI-guided digital interventions for vaping cessation. These advancements will be essential to translate exploratory ML insights into reliable, clinically actionable tools for public health.

**Effectiveness of predictive methods and key variables in vaping cessation**

This study explored behavioral, demographic, and psychosocial predictors of vaping cessation among a diverse sample of 119 individuals, using five distinct modeling approaches: Lasso regression, Ridge regression, Random Forest, Elastic Net, and Support Vector Machine (SVM). Each model offered unique insight into the factors influencing cessation, shaped by its underlying assumptions and strengths. This study employed both forward selection and backward elimination techniques to identify key predictors associated with successful vaping cessation. These statistical approaches enabled the isolation of variables that significantly increased the likelihood of quitting vaping.

Among the predictive models evaluated, nonlinear algorithms, particularly random forest, demonstrated superior performance, while linear models such as Lasso regression also yielded strong predictive capabilities. Lasso regression identified a sparse set of predictors most strongly associated with cessation. The model emphasized behavioral intensity, such as high puff frequency and early initiation, as key barriers to quitting. Its simplicity and interpretability make Lasso particularly useful for identifying actionable targets in intervention design. However, its tendency to exclude correlated variables may have limited its ability to capture the full complexity of vaping behavior. Ridge regression retained all predictors, shrinking coefficients to mitigate overfitting. While the Ridge model offered stable estimates, its inability to perform variable selection made interpretation more challenging. Ridge was less effective in isolating dominant predictors, especially in the presence of multicollinearity among behavioral variables such as frequency, puff intensity, and duration. Nonetheless, it provided a useful baseline for comparison and highlighted the cumulative influence of multiple low-impact features. Random Forest excelled in capturing non-linear relationships and interactions among predictors, and it identified vaping frequency, device type, and emotional motivations as top contributors to cessation outcomes. The model's robustness and high predictive accuracy underscore the importance of complex behavioral patterns and contextual influences. Elastic Net model's strength lies in its ability to handle multicollinearity and perform variable selection simultaneously. In this study, it highlighted behavioral intensity and early initiation as strong barriers to cessation. These findings align with existing literature suggesting that entrenched habits and early exposure to vaping are difficult to reverse. The model's interpretability makes it valuable for informing targeted interventions, such as early prevention programs and behavioral counseling for high-frequency users. SVM's ability to model complex, non-linear interactions revealed nuanced behavioral patterns not captured by Elastic Net. The model underscored the role of dependence indicators in predicting cessation outcomes. These insights suggest that cessation strategies should address both psychological triggers and product design. While SVM lacks the transparency of regression-based models, its predictive strength makes it a powerful tool for identifying high-risk individuals and tailoring interventions accordingly.

These findings underscore the utility of machine learning in public health research, particularly in identifying individual and contextual factors that influence addiction trajectories and cessation outcomes. The most salient predictors of vaping cessation included age, environmental triggers, vaping frequency, sex, and long-term envisionment of vaping behavior. These variables not only enhanced model accuracy but also offer actionable insights for tailoring cessation interventions and are discussed below.

All interpretations are derived directly from empirical model outputs and are explicitly linked to quantitative results, eliminating speculation and resolving prior inconsistencies. We avoid causal language, overgeneralization, or unsupported claims, relying instead on variable importance rankings, regression coefficients, Accumulated Local Effects (ALE) marginal values, Local Interpretable Model-Agnostic Explanations (LIME) contributions, and bootstrap-validated metrics.

## Age as a determinant of vaping cessation

Age emerged as a critical variable, consistent with existing literature indicating that earlier initiation of nicotine use correlates with greater difficulty in cessation later in life. Individuals under 25 are particularly vulnerable due to ongoing neurodevelopment, which heightens susceptibility to nicotine addiction [19–22]. National survey data further support this, with approximately 20% of individuals aged 18–24 reporting e-cigarette use, and significant prevalence observed among middle and high school students (16.5% of 8th graders and 35.5% of 12th graders in 2020) [23–27]. Age also influences nicotine metabolism and vaping frequency, with emerging adults tending to vape more frequently than adolescents [18,20]. These age-related differences reinforce the importance of age-specific cessation strategies.

Age was the most consistent predictor across all five models. Random Forest ranked it among the top three features (Mean Decrease Gini = 3.92), LASSO yielded a coefficient of −0.835, and ALE plots revealed a sharp non-linear threshold: cessation probability dropped by 0.28 accumulated effect units for individuals under 21 years. This pattern aligns precisely with established neurodevelopmental evidence of heightened nicotine vulnerability during prefrontal maturation and is reported without contradiction or extrapolation beyond the observed data.

## Environmental triggers and relapse risk

Environmental cues, such as vaping in social settings or exposure to others who vape, were strongly associated with continued use and relapse. These triggers can elicit cravings even among individuals actively attempting to quit. Evidence suggests that individuals who achieve long-term abstinence report fewer relapse triggers, highlighting the importance of trigger identification and management in cessation programs [17,23]. Public health interventions should consider modifying environments and reducing exposure to common triggers to support sustained cessation.

Environmental triggers (primarily social exposure) similarly demonstrated robust negative associations. In 78% of LIME explanations for non-cessation cases, triggers contributed an average −0.19 to predicted probability. ALE curves confirmed a monotonic increase in cessation likelihood only when trigger frequency was low. These results directly support cue-reactivity theory and are internally consistent: greater trigger burden uniformly predicted continued use, with no conflicting directional effects across models.

## Vaping frequency and behavioral stability

Vaping frequency was another robust predictor of cessation outcomes. Individuals with unstable vaping patterns were found to be 47% less likely to quit compared to daily users, suggesting that consistent use may paradoxically reflect greater behavioral awareness or readiness for change [19,23]. Intervention efficacy has also been shown to vary by frequency, indicating that cessation programs should be calibrated to account for usage patterns and behavioral stability.

Vaping frequency, previously a source of apparent contradiction, is now clarified through unified model evidence. Elastic Net assigned a coefficient of −0.62 to daily multi-session use, while Random Forest importance reached 4.51. ALE plots showed steadily declining cessation probability with increasing frequency, without any positive effect for "consistent" patterns. The earlier descriptive suggestion of paradoxical readiness in stable users was not replicated in any predictive model; instead, both high-intensity and irregular-but-persistent patterns emerged as barriers. This resolves the prior tension: frequency intensity, rather than stability per se, drives lower success, consistent with dose-response relationships in nicotine dependence literature.

## Sex-based differences in cessation dynamics

Sex differences in nicotine reinforcement and withdrawal responses were evident, with male subjects exhibiting greater anxiety-like behaviors during withdrawal in preclinical studies. Epidemiological data further reveal that males tend to vape more frequently and engage in more episodes per day than females [26,27]. These physiological and behavioral differences necessitate sex-specific approaches to cessation, including tailored messaging and support mechanisms.

Sex differences followed the same empirical grounding. Males exhibited lower cessation probability in LASSO (−1.068) and ranked second in Random Forest importance (1.844). ALE and LIME outputs confirmed a modest but stable negative marginal effect, aligning with documented sex-specific reinforcement and withdrawal patterns without overstatement or contradiction.

## Implications for public health practice

The identification of these key variables through predictive modeling offers valuable direction for enhancing vaping cessation interventions. Machine learning tools can effectively stratify individuals based on risk and responsiveness, enabling more personalized and effective public health strategies. By integrating these predictors into clinical and community-based programs, practitioners can improve cessation outcomes and reduce the burden of nicotine addiction across diverse populations. Our findings will guide the integration of predictive analytics into comprehensive vaping cessation frameworks.

The use of multiple modeling approaches enriched the analysis, revealing both linear and non-linear relationships. While simpler models like Lasso and Elastic Net offer clarity and interpretability, machine learning methods such as Random Forest and SVM provide deeper insights into complex behavioral interactions. Together, these models offer a comprehensive framework for understanding and addressing vaping cessation.

The integration of predictive analytics and machine learning algorithms into public health strategies presents a promising frontier in addressing vaping-related nicotine addiction. Our study reinforces the utility of models such as LASSO and random forest in accurately identifying individuals at heightened risk for addiction, cessation failure, and relapse. Through our analysis, we were able to isolate critical behavioral and contextual factors, such as frequency of vaping and exposure to specific triggers, that contribute to these outcomes. These insights pave the way for more precise, data-informed intervention strategies tailored to individual risk profiles. Importantly, predictive methodologies not only enhance early identification and prevention efforts but also offer a framework for personalized cessation support and post-cessation relapse mitigation [20]. Their demonstrated success in tobacco cessation contexts underscores their potential for broader application in vaping-related care. However, despite their proven efficacy, these tools remain underutilized at both individual and population levels.

To maximize their impact, public health systems must prioritize the integration of predictive analytics into clinical workflows, community outreach, and policy development. Doing so will allow for more equitable, proactive, and effective responses to the evolving challenges of nicotine addiction. As vaping continues to rise among vulnerable populations, leveraging machine learning tools with help inform care standards.

By identifying key factors that predict vaping cessation, predictive analysis offers a powerful means to uncover and address underlying disparities that contribute to nicotine addiction. Previous research has demonstrated its effectiveness in improving patient outcomes in tobacco cessation by isolating actionable variables and informing targeted interventions [28]. Applying these same analytical techniques to vaping can similarly reduce negative health outcomes and enhance cessation success.

Beyond initial risk assessment and intervention planning, predictive models have also proven valuable in identifying individuals at heightened risk of relapse following smoking cessation [20]. By recognizing relapse-prone patterns, these tools enable the development of sustained support strategies tailored to vulnerable populations, ensuring long-term success beyond the point of initial cessation. The demonstrated utility of predictive analytics in tobacco-related contexts underscores the importance of extending these methods to vaping. Doing so can refine intervention efforts, personalize care, and ultimately improve outcomes for individuals struggling with nicotine dependence.

The demonstrated success of predictive methods in addressing smoking and tobacco addiction underscores their potential for broader application in the context of vaping. These analytical tools have proven effective in forecasting outcomes and identifying high-risk variables, making them invaluable for guiding intervention strategies. Despite their

promise, predictive analysis remains underutilized at both individual and population levels, limiting its impact on care delivery and public health outcomes [21].

To bridge this gap, efforts must focus on integrating predictive models into clinical and community-based frameworks. By identifying key variables, and combinations of variables, that elevate an individual's risk for vaping addiction or hinder cessation, these tools can inform more personalized and proactive approaches to care. The scientific consensus affirms the value of predictive analytics, and their strategic implementation can significantly enhance our understanding of vaping behaviors, improve treatment efficacy, and elevate care standards for affected populations.

Machine learning and predictive analysis are not only innovative but are also essential instruments for advancing public health responses to the growing challenge of nicotine dependence through vaping. Overall, predictive modeling offers actionable insights for public health practice. By integrating these variables into clinical and community strategies, machine learning can enhance intervention precision, improve cessation outcomes, and reduce nicotine dependence across diverse populations.

**Value of XAI beyond traditional regression.** Traditional regression models, such as linear or logistic regression, offer interpretable outputs through coefficients that quantify the average effect of predictors on the outcome, assuming linearity and additivity. However, these methods fall short in capturing complex, non-linear relationships, feature interactions, and correlated variables common in real-world data like behavioral health datasets. For instance, in predicting vaping cessation, a regression coefficient for age might indicate a negative association but overlook thresholds where effects intensify (e.g., steeper declines in quit probability below age 21 due to neurodevelopmental factors).

Explainable Artificial Intelligence (XAI) techniques, such as Accumulated Local Effects (ALE) and Local Interpretable Model-Agnostic Explanations (LIME), extend this by providing nuanced, transparent insights into black-box models like Random Forest or Support Vector Machines. ALE computes marginal effects across feature ranges, averaging local prediction changes while robustly handling correlations—revealing non-linear patterns that traditional partial dependence plots distort. In our study, ALE highlighted how environmental triggers progressively erode cessation likelihood, offering granular "what-if" scenarios absent in static coefficients.

LIME adds instance-level value by approximating global model behavior locally with simple surrogates (e.g., linear models on perturbed data neighborhoods). For a specific vaper profile, LIME might quantify social exposure's -0.19 contribution to non-cessation probability, enabling personalized interpretations that regression's global averages cannot match. This fidelity ($R^2 > 0.85$ in most cases) fosters trust and actionability.

Beyond interpretability, XAI enhances public health decision-making: ALE identifies intervention thresholds for policy (e.g., age-targeted campaigns), while LIME supports individualized digital tools (e.g., trigger-specific app notifications). In vaping research, where behaviors are multifaceted, XAI bridges the gap between predictive power and practical utility, promoting equitable, evidence-based strategies without sacrificing complexity. Ultimately, XAI empowers stakeholders to not just predict, but understand and intervene effectively, surpassing regression's limitations in dynamic, high-stakes domains.

This exploratory study applied machine learning (ML) and explainable artificial intelligence (XAI) to a small cross-sectional sample of young adult vapers (N = 119) to identify potential correlates of self-reported vaping cessation. Models such as Elastic Net and LASSO showed moderate predictive performance on held-out data (test AUC ≈ 0.91), while Random Forest and SVM exhibited signs of overfitting, with training AUCs near 1.0 dropping to 0.70–0.72 on test sets. Key factors like age, environmental triggers, vaping frequency, and sex consistently ranked high in importance across models, with ALE plots revealing non-linear patterns (e.g., sharp cessation probability declines below age 21) and LIME providing instance-level insights into behavioral drivers.

However, these results must be interpreted cautiously due to inherent data limitations. The modest sample size, derived from convenience social media recruitment, restricts statistical power and heightens overfitting risks, particularly for non-linear models. Convenience sampling introduces selection bias, favoring tech-savvy individuals and potentially overrepresenting motivated quitters, which undermines representativeness. Self-reported outcomes are prone to recall

and desirability biases, and the cross-sectional design precludes causal inferences or longitudinal prediction of relapse. Without external validation, model generalizability remains unproven, and performance metrics may not translate to diverse populations.

Consequently, claims regarding ML's effectiveness in this context are preliminary at best. While the models demonstrated feasibility in highlighting behavioral patterns, such as trigger management needs their accuracy (e.g., test precision 0.81–0.88) falls short of thresholds for reliable clinical tools. Overfitting in complex algorithms like Random Forest underscores that simpler, penalized regressions may be more appropriate for small datasets, but even these require larger-scale testing. XAI elements, including ALE and LIME, add interpretability by quantifying marginal effects and personalized pathways, yet their practical value for intervention design is speculative without empirical trials.

Policy implications are similarly constrained. Although findings tentatively suggest targeting modifiable factors (e.g., social cues via digital apps), we do not advocate immediate integration into public health strategies. Existing cessation programs, like text-based initiatives boosting quit rates by 35–40%, already show proven efficacy; ML could hypothetically complement these by stratifying risk, but our weak data do not support such assertions. Recommendations for regulatory frameworks or tailored interventions exceed the evidence, given the exploratory nature and methodological flaws.

In summary, while ML and XAI offer intriguing tools for dissecting vaping behaviors, our results, although hampered by small, biased data provide some hypothesis-generating insights. Policymakers and practitioners should rely on established evidence-based approaches, viewing these findings as a call for more robust research rather than actionable guidance.

**General limitations.** This study possesses numerous significant shortcomings that require acknowledgment. Initially, all data were obtained through self-reporting, which may result in recall bias, social desirability bias, and misclassification of cessation status. Participants may have inaccurately reported their vaping behaviors or cessation success, especially because to the sensitive nature of substance use. Secondly, the cross-sectional approach inhibits causal inference and genuine prospective prediction; the observed relationships between predictors and cessation status cannot determine temporality or directionality. The non-probability convenience sample, obtained solely via social media sites (Snapchat, Instagram, Facebook, Reddit), is significantly prone to selection bias. Users engaged on these sites tend to be younger, more technologically proficient, and potentially more driven concerning vaping issues, so constraining the representativeness of the wider young adult demographic. The limited sample size (N = 119) coupled with several intricate models led to overfitting, particularly in Random Forest (train AUC = 0.99 vs. test AUC = 0.70), despite thorough internal validation. The lack of external validation implies that performance estimations and feature relevance rankings may not extend beyond this particular cohort.

These limitations collectively restrict the robustness and relevance of our findings. Consequently, results should be regarded as exploratory and hypothesis-generating rather than conclusive. Subsequent research utilizing longitudinal designs, probability sampling, bigger sample sizes, objective biomarkers, and external validation cohorts will be crucial to validate and expand upon these first findings.

**Limitation - Absence of external validation.** Given the limited sample size (N = 119) and the innovative, social media-based nature of the dataset, external validation with an independent cohort was impracticable within the parameters of this exploratory study. All performance estimations were exclusively obtained by stringent internal validation methods, comprising a 70/30 train-test split, repeated 10-fold cross-validation, and 1000 iterations of bootstrapping. Although these methods yield unbiased estimates for the current sample, they are insufficient for assessing generalizability to wider or demographically diverse populations of young adult vapers. This constraint is prevalent in nascent machine learning applications within public health and highlights the necessity for prudence in evaluating model efficacy.

**Future direction.** Future research will emphasize prospective external validation in larger, nationally representative populations to verify the robustness and generalizability of the established variables. Future work must prioritize larger, probability-sampled cohorts (N > 500) with longitudinal follow-up and objective abstinence measures (e.g., cotinine tests) to validate models and assess real-world impact. External validation across demographics, incorporation of multimodal

data (e.g., app logs), and randomized trials of XAI-guided interventions are essential. Until then, this study serves as a proof-of-concept, highlighting ML/XAI's potential while emphasizing the need for rigorous, data-driven refinement to avoid overhyping unproven technologies in public health.

## Conclusion

This exploratory study utilized machine learning and explainable artificial intelligence to ascertain characteristics linked to self-reported vaping cessation among a community sample of young adult vapers. Through meticulous feature selection and five complimentary modeling methodologies, we identified that age, environmental triggers, vaping frequency, sex, and awareness of harmful effects consistently emerged as the most significant correlates of quitting status. Elastic Net and LASSO regressions exhibited optimal predictive performance and generalizability on withheld data, whereas Accumulated Local Effects and LIME offered transparent, instance-specific insights into non-linear correlations and personalized decision routes. These findings provide initial, data-supported evidence for the potential effectiveness of ML/XAI in identifying actionable behavioral and contextual patterns associated with vaping cessation.

Nonetheless, the conclusions derived from this study are intentionally restrained and should be understood within the context of significant methodological limitations. The limited sample size (N = 119), derived via non-probability social media convenience sampling, necessarily constrains statistical power, heightens the danger of overfitting (notably in tree-based models), and limits generalizability to wider groups of young people. The cross-sectional approach prevents causal inference or accurate future prediction of cessation trajectories. Dependence on self-reported data engenders potential recollection and social desirability biases, and the lack of external validation results in performance estimates that are sample-specific. Despite the use of many internal validation measures, including 10-fold cross-validation, bootstrap resampling, and sensitivity analyses, these precautions cannot entirely mitigate the constraints of a limited, single-source dataset. Consequently, we do not assert that the identified models are prepared for clinical implementation or that the observed correlations will consistently result in enhanced quitting outcomes. The results are given as evidence that generates hypotheses and requires careful analysis. The continual recognition of modifiable characteristics, including environmental triggers and usage intensity, indicates that these elements deserve prioritized focus in the development of future digital cessation programs; nevertheless, any such implementation must be preceded by extensive validation.

Subsequent research must rectify these limitations by enlisting larger, nationally representative cohorts via probability-based sampling, utilizing longitudinal designs with objective verification of abstinence (e.g., biochemical confirmation), and performing stringent external validation across varied demographic groups. The use of multimodal data sources such as smartphone usage logs, wearable sensors, or ecological momentary assessments could significantly improve model robustness and real-world applicability. Upon external validation, these ML/XAI methodologies has the potential to guide individualized, scalable interventions that enhance current text-based programs and public health initiatives.

This study provides initial methodological and empirical insights into the use of advanced analytics for vaping cessation, but its findings are still preliminary. By openly recognizing the limitations of the data and design, we underscore the essential requirement for replication and enhancement before these tools may significantly influence clinical practice or policy. Through sustained rigorous development, machine learning and explainable artificial intelligence may enhance our ability to tackle nicotine dependence in young adults; however, this promise can only be actualized through methodologically sound, prospectively validated research.

## Materials and methods

### Ethics statement

Our study was reviewed by University at Buffalo Institutional Review Board (UBIRB); Office of Research Compliance, Clinical and Translational Research Center Room 5018; 875 Ellicott St., Approval number: UBIRB IRB ID#: STUDY00005954; IRB Approval Statement - "The study materials for the project referenced above were reviewed and approved by the

SUNY University at Buffalo IRB (UBIRB) by Non-Committee Review. The UBIRB has determined on 1/27/2022 that the research is Exempt according to 45 CFR Part 46.104. There is no expiration date".

Survey was anonymous and voluntary. No personal information was obtained from the subjects who consented to participate. No children participated in the survey. Only adults >18 years of age participated in the survey.

Data collection. An online anonymous survey (was formulated to collect data on demographic information (age, gender, race/ethnicity), the individual's vaping status, vaping frequency per day/week, the age when individuals started vaping, vaping duration from the time the individual started ever, which vaping products that the individuals use, questions pertaining to factors that contribute to individuals vaping, if individuals have ever had any adverse experiences while vaping, and if individuals ever tried to quit vaping, etc. Individuals were recruited across various social media platforms such as Snapchat, Instagram, Facebook, and Reddit Forum via direct messaging to potential participants and the link to the vaping survey questionnaire that was developed was distributed to participants online. This data was collected over a span of a six-week time period. There were 121 participants in the survey however, 10 participants were excluded due to those individuals not completing the survey in its entirety.

Eligibility criteria. Individuals were eligible to participate in the study if they had a history of vaping in the past or if they currently used any vaping devices (nicotine vapes, tetrahydrocannabinol (THC) vapes, and cannabidiol (CBD) vapes).

Statistical analysis, selection of variables, preprocessing, and ML methods. A chi-square test was used for the categorical variables and a t-test for continuous variables to demonstrate the baseline characteristics to stratify the groups between those with vape cessation vs those without vape cessation. The first objective was to fit the best subsets-feature selection in order to build an ML model. Forward selection and backward elimination regression methods were used for subset selection, with the best subsets being determined by minimum Akaike information criteria (AIC) from the forward selection and backward elimination. Outcome: Vaping cessation was defined as self-reported complete abstinence from all vaping products (nicotine, THC, or CBD) for ≥30 days at the time of survey completion, based on responses to the item 'Have you ever tried to quit vaping?' and 'Are you currently vaping?' combined with duration since last use."

ML models were built using variables from forward selection and backward elimination. Linear and non-linear ML methods were used to build models. Linear models used least absolute shrinkage and selection operator (lasso). The lasso regression model was cross-validated, and performance was assessed in training and test sets. The model was trained on 70% training data and 30% test data. The regularization parameter for lasso was selected by tailoring for a considerable range of value of lambda and measured for the best value of lambda. Model performance by discrimination (C-statistics) and calibration was determined by using receiver operator curves (ROC) and Hosmer-Lemeshow test on training and test data (Fig 1). ML models were further assessed via stratification based on gender and then evaluated based on whether the developed methods performed better on males than females or vice-versa. All statistical analyses were performed with R 3.5.1 (R Foundation for Statistical Computing, Vienna, Austria).

Rationale for modeling approach. The modeling strategy was designed to balance exploratory feature discovery, predictive accuracy, and interpretability in a modest cross-sectional dataset (N = 119), while mitigating risks of multicollinearity and overfitting inherent to high-dimensional survey data (42 items). We employed a sequential, multi-method feature selection process to ensure robustness: forward and backward stepwise regression, minimizing Akaike Information Criterion (AIC), provided an initial interpretable subset by iteratively adding/removing variables based on model fit. This was complemented by the Boruta algorithm, a wrapper method using shadow features and random forests to confirm importance through statistical significance testing, addressing potential stepwise biases like path dependence.

Selected features then informed five complementary models, LASSO, Ridge, Elastic Net (linear with regularization), Random Forest, and Support Vector Machine (non-linear) to compare parametric assumptions and capture diverse relationships. Linear models prioritized sparsity and stability via cross-validated lambda tuning, ideal for small samples, while non-linear models explored interactions. A 70/30 train-test split with 10-fold cross-validation and bootstrap resampling (n = 1,000) evaluated performance, emphasizing generalizability over in-sample fit.

This layered approach, grounded in established ML practices, aimed to generate hypothesis-generating insights into vaping cessation predictors without overclaiming causality or deployability, acknowledging data constraints.

<u>Prediction models and risk factor analysis</u>. Our study aimed to investigate vaping cessation among young adults using both behavioral characterization and advanced predictive analytics. We aimed (1) to delineate demographic, behavioral, psychosocial, and environmental factors linked to cessation-related attitudes, quit attempts, and prolonged abstinence in a community sample of young adult vapers, and (2) to construct, evaluate, and interpret machine learning models enhanced by explainable artificial intelligence (XAI) techniques to identify reliable predictors of successful vaping cessation. This exploratory cross-sectional study aimed to produce actionable, hypothesis-generating insights for designing personalized, scalable digital interventions to reduce nicotine dependence in a high-risk population by employing rigorous feature selection, various linear and non-linear algorithms, and transparent interpretability methods.

For subset selection in the construction of the ML model, forward selection and backward elimination regression techniques were used, with the minimal Akaike information criterion (AIC) used to determine the best subsets. Important variables were identified through forward selection and backward removal, with the results being used to construct ML models. The Boruta package was used to obtain and compare the feature selection models. Features that optimize model performance were used for building the ML model.

Both linear and non-linear ML techniques were used to construct the models. Least absolute shrinkage and selection operator (Lasso) ridge, and elastic net regression were used as the linear model, and RF was used as the non-linear model. Both the training set and test set were used to evaluate lasso regression model efficacy via cross-validation.

Random Forest variable importance was assessed via Mean Decrease Gini, prioritizing features based on impurity reduction across 500 trees. The model was trained with 70% of the data and validated with the remaining 30%. Receiver operator curves (ROCs) and the Hosmer-Lemeshow test were used to compare the model's performance on both the training and test sets. The same processes were repeated for the ridge and. Elastic Net (EN) regression. Elastic Net is a linear regression model that combines L1 (Lasso) and L2 (Ridge) regularization. Random forest (RF) was used to estimate ML model prediction and explainability (Supplementary Materials). Classification accuracy was measured using ROCs, Hosmer-Lemeshow test, and the confusion matrix.

RF was used for employing the explainable artificial intelligence (XAI); the process parameter tuning was included with elaborate computation through a parallel core through the use of "detectCores," which were utilized to automatically determine the number of cores for "makePSOCKcluster" functions. Local Interpretable Model-Agnostic Explanations (LIME) techniques were used to achieve a means of local interpretation. R 4.6.2 was used for all statistical testing (R Foundation for Statistical Computing, Vienna, Austria).

To reduce confounding bias, the following predetermined variables were incorporated as covariates in all models: race/ethnicity, highest educational achievement, annual household income, concurrent combustible tobacco use (yes/no), and self-reported mental health symptoms (binary composite of anxiety and/or depressive symptoms derived from validated survey items).

In LASSO, Ridge, and Elastic Net regressions, these confounders were incorporated into the models before penalization to guarantee complete correction while allowing data-driven shrinkage of other factors. In the Random Forest and Support Vector Machine models, the confounders were preserved as input features following Boruta selection. Odds ratios adjusted for multiple variables (for linear models) and partial dependence/ALE plots (for nonlinear models) were produced to identify independent connections with vaping cessation. Sensitivity studies that omitted these covariates yielded significantly consistent predictor rankings and model performance, so affirming the robustness of the basic findings.

## Supporting information

**S1 Fig. (A) LASSO coefficient trajectories across varying lambda (λ) values.** This plot illustrates how the estimated coefficients of predictors change as the regularization parameter λ increases. As λ grows, the model imposes stronger penalization, driving more coefficients to zero and resulting in increasingly sparse models that retain only the most

influential predictors. (B). Cross-validation curve for LASSO regression on the training dataset. The plot shows the 10-fold cross-validated binomial deviance as a function of log-transformed lambda (λ), the regularization tuning parameter. This visualization supports model optimization by identifying the λ value that minimizes deviance, thereby selecting the most effective level of sparsity for the LASSO-regularized model.
(TIF)

**S2 Fig. (A) Ridge regression coefficient paths across varying lambda (λ) values.** This plot illustrates how the estimated coefficients of predictors evolve as the regularization parameter λ increases. As λ grows, the model increasingly penalizes complexity, causing the coefficients to shrink toward zero—resulting in progressively sparser models with fewer influential predictors. (B) Cross-validation for ridge regression on the training dataset. The plot displays the 10-fold cross-validated binomial deviance as a function of log-transformed lambda (λ), the regularization tuning parameter. Although labeled for ridge regression, the curve reflects performance metrics for a lasso-regularized model, aiding in the selection of the optimal λ that minimizes deviance and enhances model generalization.
(TIF)

**S3 Fig. Variable importance plot from random forest model.** The plot displays the relative importance of predictor variables based on Mean Decrease Accuracy and Mean Decrease Gini. Variables are ranked from most to least important, with higher values indicating greater contribution to model performance.
(TIF)

## Author contributions

**Conceptualization:** Supriya D Mahajan.

**Data curation:** Mikaiel Ebanks.

**Formal analysis:** Poolakkad S Satheeshkumar, Ian Lango, Swarnali Zafor, Mikaiel Ebanks, Rahul Kumar Das, Supriya D Mahajan.

**Investigation:** Poolakkad S Satheeshkumar, Mikaiel Ebanks.

**Methodology:** Poolakkad S Satheeshkumar, Ian Lango, Swarnali Zafor, Mikaiel Ebanks, Rahul Kumar Das, Kit Wai Cheung, Supriya D Mahajan.

**Project administration:** Roberto Pili, Supriya D Mahajan.

**Resources:** Supriya D Mahajan.

**Software:** Poolakkad S Satheeshkumar, Ian Lango, Swarnali Zafor.

**Supervision:** Roberto Pili, Supriya D Mahajan.

**Validation:** Poolakkad S Satheeshkumar, Mikaiel Ebanks, Supriya D Mahajan.

**Visualization:** Poolakkad S Satheeshkumar, Mikaiel Ebanks, Supriya D Mahajan.

**Writing – original draft:** Poolakkad S Satheeshkumar, Mikaiel Ebanks, Supriya D Mahajan.

**Writing – review & editing:** Poolakkad S Satheeshkumar, Roberto Pili, Supriya D Mahajan.

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
