## [Decision Letter · Decision Letter 0]

9 Feb 2026

PDIG-D-25-00724Predicting Vaping Cessation in Young Adults: A Machine Learning and Explainable Artificial Intelligence (XAI) Approach to Public Health Intervention.PLOS Digital Health Dear Dr. Mahajan, Thank you for submitting your manuscript to PLOS Digital Health. After careful consideration, we feel that it has merit but does not fully meet PLOS Digital Health's publication criteria as it currently stands. Therefore, we invite you to submit a revised version of the manuscript that addresses the points raised during the review process. Please submit your revised manuscript by Apr 10 2026 11:59PM. If you will need more time than this to complete your revisions, please reply to this message or contact the journal office at digitalhealth@plos.org.  Please include the following items when submitting your revised manuscript:* A letter that responds to each point raised by the editor and reviewer(s). You should upload this letter as a separate file labeled 'Response to Reviewers'. This file does not need to include responses to any formatting updates and technical items listed in the 'Journal Requirements' section below.'. This file does not need to include responses to any formatting updates and technical items listed in the 'Journal Requirements' section below.* A marked-up copy of your manuscript that highlights changes made to the original version. You should upload this as a separate file labeled 'Revised Manuscript with Track Changes'.'.* An unmarked version of your revised paper without tracked changes. You should upload this as a separate file labeled 'Manuscript'.'. If you would like to make changes to your financial disclosure, competing interests statement, or data availability statement, please make these updates within the submission form at the time of resubmission. Guidelines for resubmitting your figure files are available below the reviewer comments at the end of this letter. We look forward to receiving your revised manuscript. Kind regards, Hadi GhasemiAcademic EditorPLOS Digital Health Laura SbaffiSection EditorPLOS Digital Health Leo Anthony CeliEditor-in-ChiefPLOS Digital Healthorcid.org/0000-0001-6712-6626 **Journal Requirements:** 1. Please provide separate figure files in .tif or .eps format. For more information about figure files please see our guidelines:   https://journals.plos.org/digitalhealth/s/figures https://journals.plos.org/digitalhealth/s/figures#loc-file-requirements   2. Please provide an Author Summary. This should appear in your manuscript between the Abstract (if applicable) and the Introduction, and should be 150–200 words long. The aim should be to make your findings accessible to a wide audience that includes both scientists and non-scientists. Sample summaries can be found on our website under Submission Guidelines:  https://journals.plos.org/digitalhealth/s/submission-guidelines#loc-parts-of-a-submission  3. We have noticed that you have uploaded Supporting Information files, but you have not included a list of legends. Please add a full list of legends for your Supporting Information files after the references list.  4. In the online submission form, you indicated that ‘Data will be available on request’.  All PLOS journals now require all data underlying the findings described in their manuscript to be freely available to other researchers, either 1. In a public repository, 2. Within the manuscript itself, or 3. Uploaded as supplementary information. This policy applies to all data except where public deposition would breach compliance with the protocol approved by your research ethics board. If your data cannot be made publicly available for ethical or legal reasons (e.g., public availability would compromise patient privacy), please explain your reasons by return email and your exemption request will be escalated to the editor for approval. Your exemption request will be handled independently and will not hold up the peer review process, but will need to be resolved should your manuscript be accepted for publication. One of the Editorial team will then be in touch if there are any issues. If the reviewer comments include a recommendation to cite specific previously published works, please review and evaluate these publications to determine whether they are relevant and should be cited. There is no requirement to cite these works unless the editor has indicated otherwise.  **Additional Editor Comments (if provided):****Reviewers' Comments:** Reviewer's Responses to Questions

**Comments to the Author**

1. Does this manuscript meet PLOS Digital Health’s publication criteria? Is the manuscript technically sound, and do the data support the conclusions? The manuscript must describe methodologically and ethically rigorous research with conclusions that are appropriately drawn based on the data presented.? Is the manuscript technically sound, and do the data support the conclusions? The manuscript must describe methodologically and ethically rigorous research with conclusions that are appropriately drawn based on the data presented.

Reviewer #1: Yes

Reviewer #2: Partly

2. Has the statistical analysis been performed appropriately and rigorously?

Reviewer #1: No

Reviewer #2: Yes

3. Have the authors made all data underlying the findings in their manuscript fully available (please refer to the Data Availability Statement at the start of the manuscript PDF file)?

The PLOS Data policy requires authors to make all data underlying the findings described in their manuscript fully available without restriction, with rare exception. The data should be provided as part of the manuscript or its supporting information, or deposited to a public repository. For example, in addition to summary statistics, the data points behind means, medians and variance measures should be available. If there are restrictions on publicly sharing data—e.g. participant privacy or use of data from a third party—those must be specified.requires authors to make all data underlying the findings described in their manuscript fully available without restriction, with rare exception. The data should be provided as part of the manuscript or its supporting information, or deposited to a public repository. For example, in addition to summary statistics, the data points behind means, medians and variance measures should be available. If there are restrictions on publicly sharing data—e.g. participant privacy or use of data from a third party—those must be specified.

Reviewer #1: No

Reviewer #2: No

4. Is the manuscript presented in an intelligible fashion and written in standard English?

Reviewer #1: Yes

Reviewer #2: Yes

5. Review Comments to the Author

Reviewer #1: attached

Reviewer #2: Review Comments to the Author

This manuscript explores the use of machine learning and explainable AI to predict vaping cessation among young adults, addressing a highly relevant public health challenge. The topic is well aligned with PLOS Digital Health, and the authors demonstrate familiarity with a broad range of ML techniques and contemporary cessation literature. Ethical approval and participant protections are appropriately described.

That said, several major issues should be addressed before the manuscript can be considered for publication:

The abstract should be rewritten in a structured format consistent with PLOS Digital Health standards. Specifically, the authors should clearly separate and label the following components: Background, Objective, Methods, Results, and Conclusions. The current abstract is narrative and overly dense, making it difficult to quickly identify the study aim, analytic approach, and key findings. A structured abstract would substantially improve clarity, accessibility, and alignment with journal expectations, particularly for a digitally focused and methodologically complex study.

Language: The manuscript is generally intelligible and written in standard academic English. However, substantial editorial refinement is needed: Repetition across the Introduction and Discussion reduces clarity and conciseness. Some sections (especially Results and Discussion) are overly long and descriptive, with limited synthesis. Terminology is occasionally inconsistent (e.g., cessation definitions, model performance claims).

Minor grammatical errors, formatting inconsistencies, and typographical issues are present.

Outcome Definition and Study Design: The primary outcome of “vaping cessation” is insufficiently defined. It is unclear whether cessation reflects a quit attempt, short-term abstinence, or sustained cessation, and over what timeframe. This ambiguity undermines interpretation of model performance and clinical relevance.

Sample Size and Model Complexity

The relatively small sample (n = 119) is a major limitation given the number of predictors and the use of complex models (Random Forest, SVM, XAI). The manuscript should explicitly discuss risks of overfitting, limited generalizability, and instability of feature importance rankings.

Statistical and ML Methodology: While multiple models are explored, the rationale for using overlapping feature-selection strategies (forward/backward selection, Boruta, LASSO) is unclear. Performance comparisons between models are sometimes inconsistent, and claims of superiority should be tempered. Calibration results indicating borderline or poor fit require discussion.

Explainability and XAI Claims: The manuscript emphasizes explainable AI, yet the practical contribution of ALE and LIME analyses to public health decision-making is not clearly articulated. The authors should clarify how these explanations add value beyond traditional regression outputs.

Data Availability and Reproducibility: The current Data Availability Statement does not meet PLOS requirements. Public deposition of de-identified data and analytic code is essential for transparency and reproducibility.

Interpretation and Scope of Conclusions: Several conclusions particularly regarding sex differences, behavioral readiness, and intervention design extend beyond what can be supported by this dataset. These interpretations should be reframed as exploratory and hypothesis-generating.

Manuscript Structure and Clarity: The manuscript would benefit from substantial condensation, especially in the Discussion, and clearer separation between results, interpretation, and implications.

This is a promising and relevant study with potential contribution to digital public health research. However, substantial revisions are required to strengthen methodological rigor, transparency, and alignment between data, analyses, and conclusions.

6. PLOS authors have the option to publish the peer review history of their article (what does this mean?). If published, this will include your full peer review and any attached files.). If published, this will include your full peer review and any attached files.

**Do you want your identity to be public for this peer review?** For information about this choice, including consent withdrawal, please see our Privacy Policy..

Reviewer #1: No

Reviewer #2: **Yes:** Andrews BaidooAndrews Baidoo

  **Figure resubmission:** While revising your submission, we strongly recommend that you use PLOS’s NAAS tool (https://ngplosjournals.pagemajik.ai/artanalysis) to test your figure files. NAAS can convert your figure files to the TIFF file type and meet basic requirements (such as print size, resolution), or provide you with a report on issues that do not meet our requirements and that NAAS cannot fix.

After uploading your figures to PLOS’s NAAS tool - https://ngplosjournals.pagemajik.ai/artanalysis, NAAS will process the files provided and display the results in the "Uploaded Files" section of the page as the processing is complete. If the uploaded figures meet our requirements (or NAAS is able to fix the files to meet our requirements), the figure will be marked as "fixed" above. If NAAS is unable to fix the files, a red "failed" label will appear above. When NAAS has confirmed that the figure files meet our requirements, please download the file via the download option, and include these NAAS processed figure files when submitting your revised manuscript. **Reproducibility:** To enhance the reproducibility of your results, we recommend that authors of applicable studies deposit laboratory protocols in protocols.io, where a protocol can be assigned its own identifier (DOI) such that it can be cited independently in the future. Additionally, PLOS ONE offers an option to publish peer-reviewed clinical study protocols. Read more information on sharing protocols at https://plos.org/protocols?utm_medium=editorial-email&utm_source=authorletters&utm_campaign=protocols

---

## [Editor Report · Decision Letter 1]

9 Apr 2026

Identifying Factors Associated with Vaping Cessation in Young Adults: A Machine Learning and XAI Approach.

PDIG-D-25-00724R1

Dear Dr Mahajan,

We are pleased to inform you that your manuscript 'Identifying Factors Associated with Vaping Cessation in Young Adults: A Machine Learning and XAI Approach.' has been provisionally accepted for publication in PLOS Digital Health.

Best regards,

Laura Sbaffi, PhD, MA, MSc

Section Editor

PLOS Digital Health